# Association of Dairy Product Consumption with Metabolic and Inflammatory Biomarkers in Adolescents: A Cross-Sectional Analysis from the LabMed Study

**DOI:** 10.3390/nu11102268

**Published:** 2019-09-21

**Authors:** Sandra Abreu, César Agostinis-Sobrinho, Rute Santos, Carla Moreira, Luís Lopes, Carla Gonçalves, José Oliveira-Santos, Eduarda Sousa-Sá, Bruno Rodrigues, Jorge Mota, Rafaela Rosário

**Affiliations:** 1Research Centre in Physical Activity, Health and Leisure, Faculty of Sport, University of Porto, 4200-450 Porto, Portugal; rutemarinasantos@hotmail.com (R.S.); carla_m_moreira@sapo.pt (C.M.); luis.iec.um@hotmail.com (L.L.); carlagoncalves.pt@gmail.com (C.G.); jomios@gmail.com (J.O.-S.); jmota@fade.up.pt (J.M.); 2Faculty of Psychology, Education and Sports, Lusófona University of Porto, 4000-098 Porto, Portugal; 3Faculty of Health Sciences, Klaipeda University, LT-91274 Klaipeda, Lithuania; cesaragostinis@hotmail.com; 4Early Start Research Institute, Faculty of Social Sciences, School of Education, University of Wollongong, Wollongong, NSW 2522, Australia; 5General Directorate of Health—National Program for Physical Activity Promotion, 1499-002 Lisbon, Portugal; 6University of Trás-os-Montes and Alto Douro, 5001-801 Vila-Real, Portugal; 7Faculty of Nutrition and Food Sciences, University of Porto, 4200-465 Porto, Portugal; 8School of Nursing, University of Minho, 4710 Braga, Portugal; edsrsa@hotmail.com (E.S.-S.); rrosario@ese.uminho.pt (R.R.); 9Interdisciplinary Centre for the Study of Human Performance, Faculty of Human Kinetic, 1499-002 Lisbon, Portugal; brunocrodrigues94@gmail.com; 10Health Sciences Research Unit: Nursing (UICISA: E), Nursing School of Coimbra (ESEnfC), 3000-232 Coimbra, Portugal

**Keywords:** dairy products, inflammation, adolescents

## Abstract

This study aimed to investigate the association between dairy product consumption and metabolic and inflammatory biomarkers in Portuguese adolescents, and whether the association differed by weight status. A cross-sectional study was conducted during the school year 2011/2012 with 412 Portuguese adolescents (52.4% girls) in 7th and 10th grade (aged 12 to 18 years old). The World Health Organization cutoffs were used to categorize adolescents as non-overweight (NW) or overweight (OW). Blood samples were collected to analyze C-reactive protein (CRP), interleukin-6 (IL-6), leptin, and adiponectin. Dairy product intake was evaluated using a food frequency questionnaire. Participants were divided by tertiles according to the amount of dairy product consumed. The associations between dairy product consumption with metabolic and inflammatory biomarkers were evaluated using generalized linear regression models with logarithmic link and gamma distribution and adjusted for potential confounders. The majority of adolescents were NW (67.2%). NW adolescents had lower IL-6, CRP, and leptin concentration than their counterparts (*p* < 0.05, for all comparisons). Higher levels of total dairy product and milk intake were inversely associated with IL-6 (P for trend <0.05, for all) in NW adolescents, but not in OW adolescents. NW adolescents in the second tertile of yogurt consumption had lower level of IL-6 compared to those in the first tertile (*p* = 0.004). Our results suggest an inverse association between total dairy product and milk intake and serum concentrations of IL-6 only among NW adolescents.

## 1. Introduction

Low-grade systemic inflammation has been associated with the development and progression of a number of chronic non-communicable diseases [1,2]. Several inflammation markers are used due to their clinical relevance, including C-reactive protein (CRP) and interleukin-6 (IL-6). IL-6 is released by various cells, such as macrophages, T and B lymphocytes, and fibroblasts, and regulate hepatic C-reactive protein (CRP) synthesis [3]. A number of studies have reported positive associations between circulating concentrations of CRP and IL-6, and the risk of cardiovascular disease [4,5], cancer [6], metabolic syndrome [7,8], and type 2 diabetes mellitus [9,10]. 

Coupled with other factors such as age, sex, weight status, physical activity, smoking, and alcohol habits, diet can influence the regulation of inflammation [11,12]. In parallel with a growing body of evidence on the role of diet, research has emerged on the association of dairy products with inflammation. It has been described that some dairy components (unsaturated fatty acids, proteins, and magnesium) may contribute to the decreased risk of developing a systemic low-grade inflammation state [13]. Conversely, a recent review of clinical trials investigating the association of dairy products with inflammation suggests that dairy product consumption, in particular fermented products, exhibited anti-inflammatory properties [14]. In addition, no adverse effect was observed for non-fermented dairy products. In line with the later review, Ulven et al. [15] concluded that milk or dairy products consumption did not exert a pro-inflammatory effect in healthy adults or adults who had an acute or chronic disease, such as overweight, obesity, metabolic syndrome, or type 2 diabetes. However, the majority of the studies analyzing the relationship between dairy products intake and inflammation were conducted with adults, whereas such studies with children and adolescents are scarce. For example, Saneei et al. [16] described in a randomized cross-over clinical trial with 60 post-pubertal girls with metabolic syndrome that consumption of the Dietary Approaches to Stop Hypertension eating pattern (i.e., rich in fruits, vegetables, whole grains, and low-fat dairy products and low in saturated fats, cholesterol, refined grains, sweets, and red meat) for 6 weeks had favorable effects on serum CRP levels compared with the usual dietary advice that consisted of healthy food choices based on the Healthy Eating Plate [17]. Conversely, a cross-sectional study suggested that serum markers of dairy fats (i.e., 15:0 and 17:0 fatty acids) are inversely related with several biomarkers such as CRP, 15-keto-dihydro-PGF2α, and 8-iso-PGF2α but only in overweight adolescents, and with IL-6 independently of participants’ weight status [18]. Indeed, adipose tissue plays an important role in the secretion of pro-inflammatory cytokines, which have been associated with the risk of adverse effects in obesity-related health conditions, promoting low-grade inflammation [19]. Excessive adipose tissue is associated with the upregulation of pro-inflamatory adipokines as leptin and decreased plasma levels of adiponectin, an anti-inflammatory adipokines [20]. It has been described that leptin associates with metabolic syndrome [21,22] and high cardiometabolic risk [23]. On the other hand, the circulating adiponectin level is decreased in the presence of insulin resistance and metabolic syndrome [22,24,25,26]. Therefore, the association between dairy products intake and low-grade inflammation among adolescents remains unclear. Accordingly, the aim of this study was to determine the association between dairy products intake and metabolic (adiponectin and leptin) and inflammatory biomarkers (CRP and IL-6) in adolescents, and to further examine if such association differs according to the weight status. In light of recent evidence, we hypothesized that dairy product intake is inversely associated with leptin and inflammatory biomarkers, and is positively associated with adiponectin, and that these associations differ by weight status.

## 2. Materials and Methods 

### 2.1. Study Design and Sample

Data for the present study derived from the Longitudinal Analysis of Biomarkers and Environmental Determinants of Physical Activity Study (LabMed Physical Activity Study), a school-based prospective cohort study carried out in five schools from the north of Portugal, aimed to evaluate the independent and combined associations of fitness levels and dietary intake on blood pressure levels of adolescents.

The LabMed Physical Activity Study methodologies have been presented elsewhere [27]. Briefly, five schools were randomly selected and the study participants’ recruitment was conducted at the selected schools. The pupils belonging to the 7th and 10th grades classes were invited to participate in the study (*n* = 1678). Baseline data was collected in the fall of 2011, for 1229 apparently healthy adolescents aged 12–14 years (7th grade) and 15–18 years (10th grade). From this sample, 534 agreed to undergo blood sampling. However, five of them were later excluded from the analysis due to CRP values > 10 mg/L, which may be indicative of acute inflammation or illness. Of these 529, 507 had complete data for dietary intake but only 412 (81.3%) had accurate reporting of energy intake. Thus, the final sample comprised 216 girls and 196 boys aged 12–18 years (mean age 7th grade: 12.7 ± 0.72 years and 10th: 15.8 ± 0.85 years).

This protocol was conducted in accordance with the Helsinki Declaration for Human Studies and approved by the Portuguese Data Protection Authority (#1112434/2011) and the Portuguese Ministry of Science and Education (0246200001/2011). Written informed consent was obtained from both participating adolescents and their parents/care-givers.

### 2.2. Anthropometric Measurements

For weight and height measurements we used a digital scale (Tanita Inner Scan BC 532, Tokyo, Japan) and a portable stadiometer (Seca 213, Hamburg, Germany) respectively. Height was measured to the nearest 0.10 cm and weight was measured to the nearest 0.10 kg. All measurements were performed with participants in light clothing, without shoes, and according to standard procedures [28]. Body mass index (BMI) was calculated from the ratio of body weight (kg) to body height (m^2^). BMI z-score was calculated using WHO AnthroPlus [29]. Participants were classified according to age- and sex-specific BMI z-score cut-off points specified by the World Health Organization as non-overweight (BMI z-score ≤ +1SD) or overweight (BMI z-score > +1SD) [30].

Body fat percentage was measured using bioelectric impedance analysis (Tanita Inner Scan BC 532, Tokyo, Japan). 

### 2.3. Pubertal Stage

Participants self-assessed their pubertal stage of secondary sex characteristics (ranging from stage I to V), according to the criteria of Tanner and Whitehouse [31]. Briefly, pubertal stage was assessed by the stage of breast development in girls and the stage of genitalia development (penis size and testicular volume) in boys (Tanner A). In addition, both of them assessed the stage of pubic hair distribution in both sexes (Tanner B). In this data, there were no individuals in the first stage of pubertal maturity. 

### 2.4. Blood Sampling

Blood samples were obtained by venipuncture from the antecubital vein in a sitting position after a 10 hour overnight fast. The samples were stored in sterile blood collection tubes in refrigerated conditions (4–8 °C), and then sent to an analytical laboratory for testing according to standardized procedures. Serum IL-6 concentrations were measured by chemiluminescence immunoassay (Immulite 2000, Diagnostic Products Corporation, Los Angeles, CA, USA); serum adiponectin and leptin concentrations by Plate Reader method (ELISA analyzer); serum high-sensitive CRP concentrations by Latex-enhanced turbidimetric assay (Siemens Advia 1600/1800, Erlangen, Germany). All assays were performed in duplicate according to the manufacturers’ instructions.

### 2.5. Cardiorespiratory Fitness 

Cardiorespiratory fitness was measured using the 20-m shuttle-run test as previously described by Léger [24]. This test required participants to run back and forth between two lines set 20 m apart. Running speed started at 8.5 km/h and increased by 0.5 km/h each minute, reaching 18.0 km/h at minute 20. Each level was announced on a tape player. The participants were told to keep up with the pace until exhausted. The test finished when the participant failed to reach the end lines concurrent with the audio signals on two consecutive occasions. Otherwise, the test ended when the subject stopped because of fatigue. Participants were encouraged to keep running as long as possible throughout the course of the test. The number of shuttles performed by each participant was recorded. Then, estimated maximum oxygen uptake (VO_2_max) was calculated by the Léger’s equation [32].

### 2.6. Socioeconomic Status

The family affluence scale (FAS) was used to assess adolescents’ socioeconomic status [33], developed specifically to measure children and adolescents’ socio-economic status in the context of the Health Behaviour in School-Aged Children Study. The FAS consists of four items on the material conditions of the household in which adolescents live. The FAS score was calculated based on the response of each item into a scale ranging from 0 to 9, with higher values meaning better socioeconomic status. 

### 2.7. Smoking

Participants self-reported their tobacco consumption, and then classified as non-smokers, former smokers (individuals who had stopped smoking for at least 6 months), occasional smokers (individuals who smoked, on average, less than one cigarette a day), and current smokers (individuals who smoked at least one cigarette a day) [34]. Occasional smokers were recoded and combined with current smokers due to the fact that they represented a small proportion of the sample (1.7%).

### 2.8. Dietary Intake

A self-administered semi-quantitative food frequency questionnaire (FFQ), validated for Portuguese adults [35], was used to measure dietary intake. This semi-quantitative FFQ was designed in accordance with criteria laid out by Willett et al. [36] and adapted to include a variety of typical Portuguese food items. The FFQ was adapted for adolescents by including foods more frequently eaten by this age group [37]; the adolescent version covered the previous 12 months and comprised ninety-one food items or beverage categories. For each item, the questionnaire offered nine frequency response options, ranging from ‘never’ to ‘six or more times per day’, and standard portion size and seasonality. Participants in a free-response section could list any foods not listed in the questionnaire. Energy and nutritional intake were estimated with regard to respondents’ ratings of the frequency, portion and seasonality of each item, using the software Food Processor Plus (ESHA Research Inc., Salem, OR, US). This program uses nutritional information from the United States that has been adapted for use with typical Portuguese foods and recipes, entering values based on the Portuguese tables of food composition [38].

We considered the consumption of total dairy product, milk (whole, reduced-fat, and fat-free), yogurt, and cheese (cottage and cream cheese). Total dairy product intake was calculated as the sum of milk, yogurt, and cheese consumed. Participants were divided according to the tertiles of the amount of each food group for the entire sample (Total dairy products: tertile 1, ≤266.1 g/day; tertile 2, 266.1–506.9 g/day; tertile 3, ≥506.9 g/day; Milk: tertile 1, ≤192.5 g/day; tertile 2, 192.5–245.0 g/day; tertile 3, ≥245.0 g/day; Yogurt: tertile 1, ≤53.6 g/day; tertile 2, 53.6–125.0 g/day; tertile 3, ≥125.0 g/day; Cheese: tertile 1, ≤4.3 g/day; tertile 2, 4.3–12.9 g/day; tertile 3, ≥12.9 g/day).

Mean adequacy ratio (MAR) was used as an indicator of nutritional quality [39]. In order to estimate MAR, nutrient adequacy ratios (NAR) of 11 micronutrients (vitamins A, B6, B12, and C, niacin, thiamin, riboflavin, folate, calcium, iron, and zinc) were calculated [40,41]. NAR was calculated as the ratio of the adolescent’s nutrient intake to the recommended dietary allowance of that nutrient for the adolescent’s age and sex (truncated at 1), using dietary reference intakes from the Institute of Medicine. MAR was calculated to a scale from 0 to 100% by following the equation: MAR = (sum of each NAR/ number of nutrients) × 100, where 100% indicates that requirements for all the nutrients were achieved. 

In this study, we only included accurate reporting of energy intake that was estimated using the Goldberg cut-off method [42] adapted by Black [43]. This approach was already described elsewhere [44]. Briefly, misreporting of energy intake was estimated using the ratio between reported energy intake and predicted basal metabolic rate to compare with 95% confidence limits (cut-offs). Cut-offs were achieved taking into account mean physical activity level, number of days of dietary assessment, within-subject coefficient of variation in energy intake, between-subject variation in physical activity, and variation in basal metabolism rate. The cut-offs achieved were 0.61 and 2.48. Misreporting of energy intake was considered when adolescents with energy intake/basal metabolic rate was below 0.61 or higher than 2.48. Therefore, from the 507 adolescents with complete data on dietary intake, 95 (19%) were excluded from the statistical analysis.

### 2.9. Statistical Analyses 

Descriptive data are presented as mean and standard deviation (SD) or median and interquartile range (25th and 75th percentiles) for continuous variables, and percentage for categorical variables stratified by non-overweight and overweight. The assumptions of normality were assessed by the Kolmogorov–Smirnov test. Independent sample t-test or the Mann–Whitney *U* test was performed to compare continuous variables between weight status, and the chi-square test was used with categorical variables. 

Generalized linear models with gamma distribution and logarithmic link function were performed to evaluate the association between each tertile of dairy products and levels of metabolic and inflammatory biomarkers, taking the lowest tertile as the reference category. Interaction between levels of inflammatory biomarkers and weight status was tested, and when interaction was statistically significant, analyses were further stratified by weight status. When no significant interaction was observed, the analysis was performed with all subjects. We adjusted models for sex, age, pubertal stage (Tanner stages A and B), total energy intake, MAR, socioeconomic status, and cardiorespiratory fitness (VO_2_max), and further for weight status, when appropriate. Bonferroni adjustment was used for multiple comparisons.

The magnitude of the associations are presented as arithmetic mean ratios and 95% confidence intervals (95% CI), which correspond to the exponentiation of the regression coefficients [45]. 

A *p*-value of < 0.05 was regarded as significant. All analyses were performed using IBM SPSS Statistics for Windows (Version 25.0. IBM Corp, Armonk, NY, USA).

## 3. Results

Descriptive characteristics of the entire sample and stratified by weight status are shown in Table 1. Prevalence of overweight was 32.8% and the majority of the participants of the study were girls (52.4%). Non-overweight subjects were older, had lower body fat percentage, and higher cardiorespiratory fitness (VO_2_max) than those who were overweight (*p* < 0.05, for all comparisons). All levels of metabolic and inflammatory biomarkers, except for adiponectin, were lower in non-overweight adolescents when compared to their counterparts (*p* < 0.05, for all comparisons). Overweight adolescents had higher percentage contributions to total energy intake of protein than non-overweight adolescents (19.5% vs. 18.5%, respectively, *p* = 0.004). There was no significant difference between non-overweight and overweight adolescents with regard to carbohydrates, fat, MAR, total dairy product, milk, yogurt, and cheese consumption.

Table 2, Table 3, Appendix A summarize the association between each dairy products group and levels of metabolic and inflammatory biomarkers. We found significant interactions between the consumption of total dairy products, milk and yogurt and CRP, IL-6 (*p* < 0.05, for all interactions) (Table 2 and Table 3). After adjusting for potential confounders, among non-overweight adolescents, but not in overweight adolescents, higher levels of total dairy product and milk intake were associated with lower levels of IL-6 (*p* for trend < 0.05, for total dairy product and milk). In non-overweight adolescents in second and third tertiles of total dairy product consumption, the mean concentration of IL-6 was, respectively, 36% and 34% lower than those in the first tertile (*p* < 0.001, for all tertiles). In non-overweight adolescents, the mean concentration of IL-6 was 37% and 35% lower in participants in the second and third tertiles of milk intake, respectively, compared to those in the first tertile (*p* < 0.001, for all tertiles) (Table 3). After adjusting for potential confounders, non-overweight adolescents belonging to the second tertile of yogurt consumption had a mean concentration of IL-6, 25% lower than those in first tertile (*p* = 0.004) (Table 3). No significant associations were found between CRP and all dairy products after adjusting for confounders (*p* > 0.05 for non-overweight and overweight adolescents) (Table 2 and Appendix A). In addition, no significant results were observed for cheese intake and IL-6, leptin, and adiponectin (Appendix A). Levels of leptin and adiponectin were not associated with the consumption of each dairy product analyzed (Appendix A).

## 4. Discussion

In this cross-sectional study, we found an inverse association between total dairy product and milk intake and serum concentrations of IL-6 in non-overweight, but not overweight adolescents. In addition, non-overweight adolescents in the middle tertile of yogurt consumption had lower levels of IL-6 than those from the lowest tertile. On the other hand, after adjustment for potential confounders no significant associations were observed between CRP serum concentration and total dairy products or milk intake in both non-overweight and overweight adolescents. Irrespective of weight status, no significant associations were found between serum concentrations of leptin and adiponectin and all dairy products analyzed. 

A systematic review of 52 clinical trials investigating inflammatory markers and its association with dairy products intake indicates that the consumption of dairy products exhibited anti-inflammatory properties in humans [14]. In that review, when results were stratified by the clinical status of the subjects, an anti-inflammatory effect of dairy products was found in healthy subjects; however, this effect was stronger in subjects with metabolic and cardiovascular disorders, including obesity and overweight. On the other hand, a systematic review of eight randomized controlled trials looking at the impact of dairy product consumption on circulating inflammatory biomarkers in overweight and obese adults suggested that dairy product consumption had no adverse effect on biomarkers of inflammation among these subjects [46]. In that review, given the methodologic factors and limitations of the selected studies, no differentiation between a beneficial or neutral impact of dairy products on inflammation could be drawn. 

Few studies have investigated the association of dairy product consumption, metabolic, and inflammatory biomarkers in adolescents, and whether weight status plays a role in such associations. In the present study, we only observed interactions between weight status and total dairy product or milk intake for CRP and IL-6. However, after adjusting for potential confounders association was only significant for serum concentration of IL-6 with total dairy products or milk intake in non-overweight adolescents. Wang et al. [18] found that weight status significantly modified the relation between serum dairy fatty acids in phospholipids (i.e., 15:0 and 17:0) and CRP, adiponectin, and markers of oxidative stress (i.e., 15-keto-dihydro-PGF2α and 8-iso-PGF2α), but not IL-6 and TNF-α. Higher levels of serum dairy fatty acids in phospholipids were associated with lower CRP, 15-keto-dihydro-PGF2α, and 8-iso-PGF2α levels only in overweight adolescents. Nevertheless, in that study, significant inverse associations were reported between IL-6 and serum dairy fatty acids in phospholipids, for all adolescents; such associations were independent of age, sex, race, pubertal stage, total energy intake, physical activity, and dietary variables. However, the majority of the studies considered weight status or BMI as confounders in statistical analysis. Data from the ATTICA study from 1514 apparently healthy men (18–87 years old) and 1528 women (18–89 years old) showed that dairy product consumption (i.e., white or yellow cheese, low-fat or full-fat milk, and yogurt) was inversely related to CRP, IL-6, and TNF-α concentrations, after adjusting for age, sex, smoking, physical activity, BMI, and dietary factors [47]. Furthermore, subjects consuming more than 14 servings of dairy products per week had 29%, 9%, and 20% lower CRP, IL-6, and TNF-α levels, respectively, when compared to those consuming fewer than 8 servings weekly. On the other hand, a cross-sectional survey carried out with 600 children aged 9–13 years showed a non-significant association between milk consumption and CRP and IL-6, after adjusting for several potential confounders, including BMI [48].

In our study we did not find linear associations for yogurt intake and cheese intake with inflammatory biomarkers. However, lower levels of IL-6 were found in the second tertile of yogurt consumption among non-overweight adolescents. These results remain statistically significant after adjusting for potential confounders. Likewise, Gadotti. et al. [49], in a cross-sectional analysis among a representative sample of Brazilian adults from São Paulo City, showed that subjects in the highest tertile of yogurt consumption had a lower odds for inflammatory status than those in the lowest tertile (odds ratio = 0.34, 95% CI: 0.14–0.81). Results from the Brazilian study also indicated that increasing yogurt consumption might have an inversely linear association with inflammatory status. On the contrary, a higher odds ratio was found for the second tertile of cheese consumption compared to first one (odds ratio = 2.49, 95% CI: 1.09–5.75). In our study, there was no evidence that higher intake of cheese was associated with higher levels of inflammatory biomarkers and leptin, or lower adiponectin. Moreover, no significant results were observed for the relation of total dairy product, milk and yogurt consumption, and adipokines serum concentrations. Likewise, González et al. [50] did not find any associations between the intake of cheese, yogurt and fermented milk, and serum levels of leptin, among other health-related biomarkers, in healthy adults. 

Some studies have investigated the possible mechanisms underlying the anti-inflammatory effects of dairy products, remaining controversial which dairy nutrients modulate the inflammatory process. Overall, it has been suggested that specific dairy fats and proteins may reduce inflammatory biomarkers by inhibiting nuclear factor κB [51,52,53] and modulating gut microbiota [54]. Data from a cross-sectional study with 130 healthy adults found that fermented dairy products (i.e., yogurt, cheese, and fermented milk) modulate the fecal microbiota [50]. In particular, yogurt seems to have higher ability to modulate intestinal microbiota, while cheese intake showed a significant effect in fecal short chain fatty acids concentration. Thereby, dairy products contain a large variety of nutrients, which may impact inflammatory markers differently; thus, further studies are warranted on the effect of dairy nutrients in inflammation regulation.

Our study is not without limitations. First, given that this study has cross-sectional design, any conclusions related to cause and effect cannot be drawn. Second, potential limitations that may advert from the use of self-dietary intake data should also be considered. Yet, the FFQ used in our study has been previously tested [55], and we only considered in our analysis accurate reporting of energy intake. Moreover, we cannot support that any kind of milk is protective against high levels of IL-6, since we considered whole, reduced-fat, and fat-free milk altogether in the analysis. However, the majority of adolescents (92.4%) consumed reduced-fat and fat-free milk. In addition, anti-inflammatory activity has been described for both low-fat and high-fat dairy products [14]. Third, we cannot rule out the presence of residual confounding; however, we accounted for several factors, such as age, sex, pubertal maturity, cardiorespiratory fitness, and dietary factors that may influence both dairy consumption and metabolic and inflammation biomarkers concentrations. In particular, previous studies have suggested significant associations between cardiorespiratory fitness and inflammatory biomarkers [56] or adipokines [57]. Finally, we cannot exclude that interactions and associations between the consumption of total dairy products, milk, and yogurt and inflammatory biomarkers may have occurred by chance, however even after controlling for confounders results are in line with formulated hypothesis.

In summary, we found a negative association between total dairy product and milk intake with serum levels of IL-6 only in non-overweight adolescents. In addition, yogurt consumption was associated with lower serum levels of IL-6 in non-overweight adolescents. Evidence from prospective and randomized clinical studies is warranted to examine the effects of long-term consumption of dairy products and its sub-groups on low-grade inflammation.

## Figures and Tables

**Table 1 nutrients-11-02268-t001:** Sample characteristics for total sample and by weight status.

	Total (*n* = 412)	Non-Overweight (*n* = 277)	Overweight (*n* = 135)	*p* *
Age, years				
7th grade	12.7 ± 0.72	12.8 ± 0.74	12.7 ± 0.70	0.536
10th grade	15.8 ± 0.85	15.9 ± 0.89	15.6 ± 0.69	0.021
Sex, *n* (%) of girls	216 (52.4)	149 (53.8)	67 (49.6)	0.427
Socioeconomic status (FAS)	6.4 ± 1.65	6.4 ± 1.71	6.5 ± 1.51	0.471
Smoking status, *n* (%)				
Non-smoker	369 (89.6)	249 (89.9)	120 (88.9)	0.260
Former smoker	32 (7.8)	23 (5.6)	9 (2.2)	
Current smoker	11 (2.7)	5 (1.2)	6 (1.5)	
Pubertal stage A, *n* (%)				
II	30 (7.3)	21 (7.6)	9 (6.7)	0.415
III	136 (33.0)	88 (31.8)	48 (35.6)	
IV	194 (47.1)	128 (46.2)	66 (48.9)	
V	52 (12.6)	40 (14.4)	12 (8.9)	
Pubertal stage B, *n* (%)				
II	29 (7.0)	17 (6.1)	12 (8.9)	0.279
III	85 (20.6)	53 (19.1)	32 (23.7)	
IV	207 (50.2)	148 (53.4)	59 (43.7)	
V	91 (22.1)	59 (21.3)	32 (23.7)	
Body fat, %	21.1 ± 8.50	17.7 ± 6.53	28.2 ± 7.68	<0.001
VO_2_max (mL/kg/min)	42.0 ± 6.74	43.2 ± 6.97	39.6 ± 5.49	<0.001
Inflammatory biomarkers				
CRP, mg/L	0.20 (0.11; 0.77)	0.13 (0.11; 0.51)	0.46 (0.18; 1.60)	<0.001
IL-6, ng/L	1.90 (1.90; 3.40)	1.90 (1.90; 2.95)	1.90 (1.90; 3.90)	0.033
Adipokines				
Adiponectin, mg/L	10.4 (7.7; 14.5)	10.8 (8.3; 15.1)	9.1 (7.2; 13.4)	0.002
Leptin, ng/mL	2.80 (0.90; 5.90)	1.60 (0.60; 4.15)	5.90 (3.10; 10.70)	<0.001
Total energy intake, kcal/day	2127.7 ± 680.98	2159.0 ± 692.06	2063.5 ± 655.51	0.182
Carbohydrate, % of energy	49.9 ± 7.02	50.0 ± 7.08	49.6 ± 6.91	0.567
Protein, % of energy	19.0 ± 3.60	18.6 ± 3.42	19.7 ± 3.87	0.004
Total fat, % of energy	32.4 ± 4.67	32.7 ± 4.75	31.9 ± 4.48	0.124
MAR, % adequacy/day	96.3 (92.4; 98.9)	96.3 (92.7; 99.0)	95.8 (91.7; 98.6)	0.828
Total dairy products, g/day	324.4 (244.0; 629.1)	346.7 (244.0; 632.9)	310.4 (244.0; 569.4)	0.310
Milk, g/day	244.0 (191.7; 610.0)	244.0 (191.7; 610.0)	244.0 (191.7; 261.3)	0.414
Yogurt, g/day	53.6 (17.9; 125.0)	53.6 (17.9; 125.0)	53.6 (17.9; 125.0)	0.817
Cheese, g/day	12.9 (2.0; 23.6)	12.9 (2.0; 23.6)	4.3 (2.0; 12.9)	0.822

Values are mean ± standard deviation, median (P25; P75) or absolute frequency (relative frequency). CRP, C-reactive protein, IL-6, Interleukin-6; FAS, family affluence scale; MAR, mean adequacy ratio. * *p*-value was calculated based on chi-square test for categorical variables, independent t-test (socioeconomic status, body fat, VO_2_max, total energy intake, macronutrients intake) or Mann–Whitney *U* test (all inflammatory biomarkers, MAR, all dairy products group) for continuous variables.

**Table 2 nutrients-11-02268-t002:** Association of total dairy products, milk and yogurt intake with C-reactive protein concentration.

	C-Reactive Protein ^a, b, c^							
	Non-Overweight				Overweight			
	Model 1		Model 2		Model 1		Model 2	
	AMR (95% CI)	*p*-Value	AMR (95% CI)	*p*-Value	AMR (95% CI)	*p*-Value	AMR (95% CI)	*p*-Value
**Tertiles of total dairy products intake**								
T1	Reference *		Reference		Reference		Reference	
T2	0.98 (0.67; 1.44)	0.926	0.98 (0.66; 1.45)	0.921	1.52 (0.95; 2.44)	0.080	1.23 (0.73; 2.08)	0.430
T3	0.51 (0.35; 0.74)	<0.001	0.80 (0.53; 1.20)	0.284	1.30 (0.78; 2.17)	0.306	1.30 (0.68; 2.51)	0.427
**Tertiles of milk intake**								
T1	Reference		Reference		Reference		Reference	
T2	1.18 (0.79; 1.76)	0.414	1.16 (0.77; 1.75)	0.480	1.50 (0.90; 2.50)	0.117	1.19 (0.71; 2.00)	0.507
T3	0.61 (0.40; 0.92)	0.019	1.02 (0.65; 1.59)	0.932	1.49 (0.83; 2.66)	0.180	1.49 (0.74; 3.03)	0.267
**Tertiles of yogurt intake**								
T1	Reference		Reference		Reference		Reference	
T2	1.40 (0.99; 1.99)	0.058	1.33 (0.96; 1.84)	0.083	1.24 (0.77; 1.99)	0.375	1.30 (0.80; 2.10)	0.294
T3	0.53 (0.34; 1.01)	0.055	0.96 (0.57; 1.65)	0.910	1.49 (0.69; 3.20)	0.308	1.39 (0.61; 3.19)	0.432

AMR, arithmetic mean ratio; T, tertile. ^a^
*P*_interaction_ [total dairy products × weight status] = 0.039; ^b^
*P*_interaction_ [milk × weight status] = 0.041; ^c^
*P*_interaction_ [yogurt × weight status] = 0.012. ** p*_for trend_ = 0.003. Total dairy products: T1, ≤266.1 g/day; T2, 266.1–506.9 g/day; T3, ≥506.9 g/day; Milk: T1, ≤192.5 g/day; T2, 192.5–245.0 g/day; T3, ≥245.0 g/day; Yogurt: T1, ≤53.6 g/day; T2, 53.6–125.0 g/day; T3, ≥125.0 g/day. Model 1: Unadjusted model; Model 2: Adjusted for age, sex, Tanner stage A and B, total energy intake, mean adequacy ratio, socioeconomic status, and cardiorespiratory fitness (VO_2_max).

**Table 3 nutrients-11-02268-t003:** Association of total dairy products, milk and yogurt intake with interleukin-6 concentration.

	Interleukin-6 ^a, b, c^							
	Non-Overweight				Overweight			
	Model 1		Model 2		Model 1		Model 2	
	AMR (95% CI)	*p*-Value	AMR (95% CI)	*p*-Value	AMR (95% CI)	*p*-Value	AMR (95% CI)	*p*-Value
**Tertiles of total dairy products intake**								
T1	Reference *		Reference ^†^		Reference		Reference	
T2	0.60 (0.49; 0.75)	<0.001	0.64 (0.52; 0.80)	<0.001	0.91 (0.67; 1.24)	0.542	1.05 (0.76; 1.45)	0.774
T3	0.67 (0.54; 0.82)	<0.001	0.66 (0.53; 0.82)	<0.001	1.02 (0.73; 1.43)	0.902	1.28 (0.91; 1.81)	0.156
**Tertiles of milk intake**								
T1	Reference ^†^		Reference ^‡^		Reference		Reference	
T2	0.60 (0.48; 0.75)	<0.001	0.63 (0.51; 0.78)	<0.001	0.96 (0.69; 1.33)	0.792	0.94 (0.64; 1.31)	0.719
T3	0.64 (0.51; 0.80)	<0.001	0.65 (0.51; 0.82)	<0.001	0.76 (0.52; 1.10)	0.148	0.91 (0.62; 1.33)	0.616
**Tertiles of yogurt intake**								
T1	Reference		Reference		Reference		Reference	
T2	0.73 (0.65; 1.14)	0.001	0.75 (0.61; 0.91)	0.004	0.69 (0.51; 0.93)	0.014	0.75 (0.54; 1.01)	0.056
T3	0.84 (0.63; 1.14)	0.278	0.87 (0.64; 1.19)	0.377	1.41 (0.88; 2.29)	0.156	1.50 (0.94; 2.39)	0.0.86

AMR, arithmetic mean ratio; T, tertile. ^a^
*P*_interaction_ [total dairy products × weight status] = 0.039; ^b^
*P*_interaction_ [milk × weight status] = 0.041; ^c^
*P*_interaction_ [yogurt × weight status] = 0.041. ** p*_for trend_ = 0.003; ^†^
*p*_for trend_ = 0.001; ^‡^
*p*_for trend_ = 0.002. Total dairy products: T1, ≤266.1 g/day; T2, 266.1–506.9 g/day; T3, ≥506.9 g/day; Milk: T1, ≤192.5 g/day; T2, 192.5–245.0 g/day; T3, ≥245.0 g/day; Yogurt: T1, ≤53.6 g/day; T2, 53.6–125.0 g/day; T3, ≥125.0 g/day. Model 1: Unadjusted model; Model 2: Adjusted for age, sex, Tanner stage A and B, total energy intake, mean adequacy ratio, socioeconomic status, and cardiorespiratory fitness (VO_2_max).

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
