# Peer review of "Association of Dairy Product Consumption with Metabolic and Inflammatory Biomarkers in Adolescents: A Cross-Sectional Analysis from the LabMed Study"

_nutrients, 2019, doi:10.3390/nu11102268_

Round 1

Reviewer 1 Report

After study results on the relevance of sub-chronic inflammation in the development of NCDs associated with overweight and obesity are accumulating, the study at hand can be considered as relevant and of interest for the research community. The study design is appropriate and the statistical evaluation is fine, but deserves a little bit more critical reflection. The language is clear and goal-leading.

Minor:

Abstract:

55+56: Unclear what “for all” means

Information should be given that the adolescents were coming from two classes (7th and 10th grade).

Introduction:

l.66: Please be more precise what “multiple serious conditions” means.

l. 68: Remove “The” from the start of the sentence.

Somewhere in the part between lines 71 and 77, it should be mentioned that the transcription of CRP is regulated by the IL-6 receptor and that adipose tissue is the source of leptin and adiponectin. It should be made clear that there is a considerable difference between immunoregulating cytokines (IL-6, produced primarily in leukocytes) and the adipokines.

Material and methods:

l. 121: Providing the information that the mean age was 14.4±1.72 years generates the impression that the majority of the adolescents was around 14 years, but in reality we have a bimodal distribution. Means and averages should only be used for more or less normally distributed datasets, which does not seem to be the case here. I would suggest to provide two averages with SDs: one for the 7th-graders and one for the 10th-graders. Applies also to Table 1.

l. 132-134. Did the authors use a software package (AnthroPlus or similar) to calculate the sex- and age-dependent Z-scores? If so, this should be mentioned here.

l. 179: Remove “the”.

l. 188: How was the “Food Processor Plus” adapted to Portuguese foods? Reference?

l. 201: What does “truncated at 1” mean? One digit behind the decimal separator? Or that all values exceeding 100% were set to 100% for the NAR/MAR calculation? Please clarify.

l. 216/217: So, interquartile ranges were used? Please mention that.

For the group comparisons of ratio values (Student’s t test), did the authors check the prerequisite of homogeneity of variances?

l. 218/219: How did the authors proceed when the Kolmogorov–Smirnov test did not indicate normal distribution?

l. 231: Replace “that” by “which”.

Results

l. 235: Change “all” to “entire”.

l. 238, 240 and other places: clarify what “all” in “for all” means.

l. 243: Unclear what “other dietary variables” means – this is not adequately characterized in the Material and methods section.

l. 244/Table 1: I completely do trust the authors with respect to the quality of their data; nevertheless, I’d like to ask them to re-check the values on milk consumption: “244.0 (191.7; 610.0); 244.0 (191.7; 261.3)” for the groups NW/OW. I just wonder how can it be that the average value is identical to 4 digits but that the SD ranges differ noteworthy, one being considerably more skewed than the other?

On which basis did the authors decide to report mean±standard deviation or median (P25; P75)? Do they report both? Typically, mean±standard deviation is used when parametric tests (Student’s) are used and median (P25; P75) is used in case of non-parametric tests (Mann-Whitney-U etc.).

All tables showing no statistical differences (Tables 4 and 5) should be moved to the supplementary material. Their information content is minimal (“no difference”) and they take a lot of space.

Discussion

l. 311+313+315: Change “phospholipids dairy fatty acids” to “dairy fatty acids in phospholipids”.

l.346: Angiotensin-converting enzyme is rather related to blood pressure, not inflammation.

I miss in the Discussion an attempt to explain why the differences (i.e. those occurring in IL-6 levels) occur in the non-overweight part of the study group. Any publications on this aspect?

In the Limitations section (l. 353-365), the authors should estimate how many significant differences they found given the fact that they performed at least 109 statistical comparisons. With that high number of comparisons, the occurrence of a few “statistically significant” differences is likely to occur by chance. In 2 brief sentences, the authors may want to describe the chance for the occurrence of these “artificial” significances” and why the pattern by which the significances occurs points towards a real association between e.g. IL-6 levels and intakes of different kinds of milk products and milk.

Reviewer 2 Report

The paper “Association of dairy product consumption with metabolic and inflammatory biomarkers in adolescents: a cross-sectional analysis from the LabMed study) describes levels of 4 inflammatory/metabolic markers in relation to relative body weight and dairy product intake in adolescents.

Overall the paper is well written and the study group likely large enough to capture a reasonable difference. There are, however, some major and minor comments and questions that emerge while reading the paper.

The first major comment refers to that the structure of the paper does not follow the title and the first sentence in the abstract. The main text starts with body weight and all sections tend to give priority to weight over dairy products. The paper would flow and catch the reader better if restructured. The second major comment refers to how tertile allocation was done (see below).

The minor comments (apart from the tertile comment Line 192) are:

The abstract reads well but it would be helpful to already in the abstract see to proportions NW and OW.

Introduction: as indicated above I suggest the introduction to start with a paragraph on dairy products and diet.

In the description of the present knowledge of dairy products and inflammation and health outcomes the authors refer to studies that report from a health point positive studies. I think they should balance this by mentioning some studies that indicate that especially non-fermented milk is not non-controversial from this point of view, i.e. see indices for diet inflammation potential.

Line 112. How about adding a flow chart?

Line 121: please give proportions of the two age groups.

Line 126. Do the authors mean care-givers instead of tutors?

Line 142: public should likely be pubic

Line 145: were the samples stored and sent as whole blood, i.e. without centrifugation and serum separation? If so, may variations in storing time and blood cell integrity have affected samples differently?

Line 154: inclusion of measures should be substantiated by the targeted research question and not on availability in a basic study where it is nested. Thus, how is the inclusion of Cardiorespiratory fitness measurement substantiated in relation to this research question on dairy products and selected biomarkers.

Line 190. Please provide a more precise description on the dairy questions, i.e. number per type of product.

Line 192. How was ranking for tertile allocation made. This is a major question since there are likely systematic differences in intake between boys and girls and the two age groups with the obvious risk that the youngest girls are in the lowest tertile and the oldest boys in the top tertile.

Line 213. It should be clarified that these 95 were already excluded among the final numbers of 412 participants. Here a reference to a flow chart would be very useful?

Line 227: The authors write: Adjusted models were included as confounders sex, age, pubertal stage (Tanner stages A and B), total energy intake, MAR and cardiorespiratory fitness (VO2max), and further for weight status, when appropriate.”First, the sentence does not read well. Secondly, socio-economic status and smoking were scored and I would assume these measures would be more appropriate to include as potential confounders than cardiorespiratory fitness both from a diet recording accuracy and life condition point of view. Thirdly, were CRP values transformed before entering the model?

No line specification: I cannot find a definition of what p-value the authors consider statistically significant and if they see any need for adjusting for multiple comparisons.

Table 1. I am surprised to see exactly the same numbers to the first decimal for the dairy products in two groups. Is this correct? Check the footnote for typos, i.e. “Qui-squared test”

Tables 2- 5. I find it impossible to assess the results presented in the Tables when the ranking is unclear and especially since some of the results were unexpected to me at least. Further, I am not familiar with the term “arithmetic mean ratio” and it was not described in the statistics section. Please clarify.

Line 288: I do not think that the results as presented in the tables are correctly summarized in the first paragraph of the discussion. I see several statistically significant p-values given that p<0.05 are considered. “In this cross-sectional study, we found an inverse association between total dairy product and milk intake and serum concentrations of IL-6 in non-overweight, but not overweight adolescents. No significant associations were observed between CRP serum concentration and total dairy products or milk intake in both non-overweight and overweight adolescents. Irrespective of weight status, no significant associations were found between serum concentrations of leptin and adiponectin and all dairy products analysed.”

Line 366. Please avoid the term negatively here to avoid confusion between biological and statistical negative association.

Round 2

Reviewer 2 Report

The paper has improved significantly and the authors responses are adequate. However, there are a few typing errors that have slipped into the new text.

Line 86 products

Line 90 However, the majority of the studies analyzing the relationship between dairy products 89 intake and inflammation were conducted with adults, whereas such studies

Line 507 Should this not be Chi-square and not Qui-square?

Line 721 should read “On the other hand

Line 804 Finally, we cannot exclude that interactions and associations between the consumption of total dairy products, milk and yogurt and inflammatory markers may have occurred by chance, however even after controlling for confounders results are in line with 806 formulated hypothesis.

As for the tertile allocation issue, I agree with the authors that the table they show indicate that they do not have a skewness in there tertile proportions and I accept it, but still the table they need to show is a cross tabulation for the proportion sex * age group split by tertile group. If these proportions are similar than ranking among all seem appropriates.
